# Redox Signaling and Sarcopenia: Searching for the Primary Suspect

**DOI:** 10.3390/ijms22169045

**Published:** 2021-08-22

**Authors:** Nicholas A. Foreman, Anton S. Hesse, Li Li Ji

**Affiliations:** Laboratory of Physiological Hygiene and Exercise Science, School of Kinesiology, College of Education and Human Development, University of Minnesota, 1900 University Ave, Minneapolis, MN 55455, USA; forem056@umn.edu (N.A.F.); hesse151@umn.edu (A.S.H.)

**Keywords:** aging, mitochondria, redox signaling, sarcopenia, skeletal muscle, peroxiredoxin

## Abstract

Sarcopenia, the age-related decline in muscle mass and function, derives from multiple etiological mechanisms. Accumulative research suggests that reactive oxygen species (ROS) generation plays a critical role in the development of this pathophysiological disorder. In this communication, we review the various signaling pathways that control muscle metabolic and functional integrity such as protein turnover, cell death and regeneration, inflammation, organismic damage, and metabolic functions. Although no single pathway can be identified as the most crucial factor that causes sarcopenia, age-associated dysregulation of redox signaling appears to underlie many deteriorations at physiological, subcellular, and molecular levels. Furthermore, discord of mitochondrial homeostasis with aging affects most observed problems and requires our attention. The search for the primary suspect of the fundamental mechanism for sarcopenia will likely take more intense research for the secret of this health hazard to the elderly to be unlocked.

## 1. Introduction

Sarcopenia is defined as the loss of mass and strength of skeletal muscle, largely as a result of aging [1]. Sarcopenia negatively affects quality of life and is associated with failure of independent living and early mortality. In addition, the prevalence of sarcopenia is expected to increase as the world’s population ages [1]. This health problem brings an enormous financial burden to society: those with sarcopenia incur an additional cost of $2316 per person per year compared to those without sarcopenia [2]. In addition, the estimated cost of hospitalization for those with sarcopenia is $40.4 billion annually [2]. Therefore, understanding the mechanisms that underlie sarcopenia may lead to therapies and drugs that mitigate these negative impacts. Unfortunately, despite decades of research, the exact mechanisms that underlie sarcopenia have yet to be delineated.

Among all the potential etiological foundations of sarcopenia, the generation of reactive oxygen species (ROS), with the associated oxidative damage and/or defective redox signaling, has stood out as a viable explanation [3,4,5]. With age, cells produce more ROS, even at the resting state, primarily from the mitochondria (mtROS) and from NADPH oxidase (NOX) [3,4]. Although antioxidant enzyme activities in muscle increase with age [5], this compensatory adaptation does not completely counteract the rising oxidative stress. Deleterious oxidative modification of macromolecules causes not only multiple cellular dysfunctions but also distortion of signal transduction pathways that control protein turnover, mitochondrial homeostasis, energy metabolism, antioxidant gene expression, and redox balance (Figure 1). In this short communication, we provide an update and critical review on how increasing ROS with age may modulate the various contributing factors in the development of sarcopenia, such as impaired protein turnover, mitochondrial dysfunction, improper 5′ adenosine monophosphate-activated protein kinase (AMPK) signaling, apoptosis and regeneration, inflammation, neuromuscular dysfunction, and nicotinamide adenine dinucleotide (NAD^+^) depletion. Finally, we focus on the effort to search for a potential key link between impaired redox signaling and functional deterioration in sarcopenic muscle and highlight the recent discoveries about the role of peroxiredoxin (Prx), a family of antioxidant enzymes that respond to peroxide levels and thereby mediate signal transduction in mammalian cells [6].

## 2. Protein Turnover

The quantity of muscle mass is determined by the rates of protein synthesis and degradation. With age, protein synthesis decreases while degradation increases, tipping the balance towards atrophy [7].

### 2.1. Protein Degradation

Muscle protein degradation is governed by four major proteolytic pathways: the ubiquitin–proteasome system (UPS), the calpains, the caspases, and the autophagy-lysosomal pathway [7]. Two of the most important E3 ubiquitin ligases with respect to UPS are Muscle RING-finger protein-1 (MuRF-1) and atrophy gene-1/muscle atrophy f-box (Atrogin-1/MAFbx). These proteins are responsible for the degradation of several myofibrillar, sarcomeric, and related regulatory proteins [8,9]. The UPS is commonly upregulated in many other wasting conditions and is heavily influenced by the forkhead box class O (FOXO) transcription factor family [10]. In sarcopenia, however, upregulation of the UPS and the influence of FOXO likely play a minor role [7,11]. With some exceptions [12], the FOXO pathway, including MuRF-1 and Atrogin-1, does not change or is even downregulated with age [1]. Instead, the calpains and autophagy pathways are thought to exert a larger influence during sarcopenia [7].

The calpain (calcium-dependent, non-lysosomal cysteine protease) system is a calcium-dependent ATP-independent pathway that cleaves myofibril proteins [13]. Calpain content and activity are known to increase with age [14], and H_2_O_2_ has been shown to increase calpain 1 and 2 expression and activity in C2C12 myotubes and in human myoblasts [15,16]. Thus, age-related increases in H_2_O_2_ may upregulate calpain-mediated degradation. The mechanism of age-related activation of the calpain system by ROS may be in part due to a higher intracellular Ca^2+^ concentration. Previous research suggests that oxidative damage induces a “leaky” ryanodine receptor [17], which impairs Ca^2+^ reuptake via sarcoplasmic reticulum Ca^2+^-ATPase (SERCA) pumps [18,19,20], thus leading to a higher intracellular Ca^2+^ concentration [21]. Additionally, ROS can increase the susceptibility of protein degradation by calpains [22], because oxidative modification alters protein secondary and tertiary structure [23]. In addition, oxidized proteins are more readily degraded by the UPS because they are more easily ubiquitinated [21,24]. Furthermore, some proteosomes are known to degrade oxidized proteins without ubiquitination [25]. This evidence suggests that elevated ROS in sarcopenic muscles triggers excessive calpain proteolysis.

Caspases are cysteine-dependent, aspartate-targeting proteases that play an important role mainly in programmed cell death and have been investigated within the context of sarcopenia [13]. Caspases, particularly caspase-3, increases with H_2_O_2_ in C2C12 myotubes [26]. Inhibition of caspase-3 has been shown to reduce cleavage of MyHC-2 and limit a calpain-2–caspase-3 interaction that degrades both α-actinin and MyHC-2 [14]. Despite this, the proteolytic role of caspases in sarcopenia is likely minor. With age, nNOS activity is reduced; this limits S-nitrosylation, which impairs calpain activity. However, transgenic nNOS expression to limit the decline in S-nitrosylation with age did not change caspase-3-dependent proteolysis [14]. It seems the role of caspases in sarcopenia is more important with respect to apoptosis, as is discussed later.

### 2.2. Protein Synthesis

The most well-known pathway for muscle protein synthesis is the PI3K/Atk(PKB)/mechanistic target of rapamycin (mTOR) pathway. Details of this pathway can be found in many previous reviews [10,13,27]. This pathway is stimulated by insulin, growth factors, mechanical loading, and feeding [27]. In addition, Akt inhibits FOXO activity, thereby reducing activation of proteolysis and autophagy [13]. Research on the activation of this pathway has revealed inconsistent results regarding the changes in mTOR pathway with age and its impact on muscle protein levels [28]. Basal levels of protein synthesis are not marred with age, but the response of protein synthesis machinery to anabolic stimuli is blunted in a phenomenon known as anabolic resistance [29]. For example, aged subjects show impaired Akt/mTOR/p70S6K signaling after muscle contraction [28], by nutritional interventions [30], and after insulin receptor binding [31].

ROS and inflammation may contribute to anabolic resistance at several levels in sarcopenia. Elevated levels of ROS such as H_2_O_2_ can inhibit phosphorylation of Akt, mTOR, and the downstream mTOR targets 4E-BP1 and p70S6K [13,21]. Excess ROS produced in aged muscles may inhibit key components of the Akt/mTOR pathway, thereby limiting their capacity to respond to exercise stimuli. Indeed, reducing oxidative stress in aging can improve exercise adaptation. For example, antioxidant supplementation was reported to restore age-related defective leucine stimulation on protein synthesis in rats [32]. Unfortunately, antioxidant supplementation alone may be insufficient to maintain muscle mass, as this approach was not shown to be successful in two human studies [33,34].

The above evidence demonstrates that oxidative stress from excessive ROS is capable of blunting protein synthesis during sarcopenia. However, ROS levels alone may not be sufficient to determine the effect on protein synthesis. Experiments in C2C12 cells by Tan et al. [35] showed that the effect of ROS on Akt phosphorylation depended on the balance between thiol oxidation of Akt, phosphatase and tensin homolog (PTEN), and protein phosphatase 2A (PP2A), as both PTEN and PP2A can dephosphorylate and suppress Akt signaling. The authors reported greater Akt phosphorylation when Akt, PTEN, and PP2A were all oxidized, whereas Akt phosphorylation was diminished when Akt alone was oxidized. It was concluded that Akt was the enzyme most sensitive to ROS-induced thiol oxidation, and low but chronic oxidative stress may lead to impaired protein synthesis. 

Ironically, although anabolic resistance presents an issue in sarcopenia, constitutive activation of the Akt/mTOR pathway promotes sarcopenia [11,36,37]. Sandri et al. [11] reported that Akt overexpression accelerated a sarcopenic phenotype in mice, demonstrating the importance of protein degradation to preserve muscle quality. Newer research bolsters this idea by using rapamycin to delay sarcopenia in mice with constitutively active mTORC1 [36]. The same study reported that atrophy induced by adding inflammatory cytokines to C2C12 cells was blunted with rapamycin treatment. It seems the elevated mTORC1 signaling in sarcopenia could be due to excessive oxidation of the phosphatases that normally provide a check on this pathway [35].

Although muscular adaptation to anabolic stimuli decreases in the elderly as previously mentioned, exercise is proven to improve muscle mass and function [38]. Both animal and human research indicates that aged skeletal muscle maintains sensitivity to exercise-induced redox signaling. Endurance training promotes mitochondrial biogenesis mainly through the upregulation of the PGC-1α signaling pathway [39]. Resistance training in the elderly can improve muscle strength and mass by favorably altering autophagy, apoptosis, and potentially muscle protein synthesis. These changes are likely mediated by changes to the IGF-1/Akt/mTOR and Akt/FOXO pathways, but more research is necessary to elucidate these mechanisms [40].

Another powerful intervention of sarcopenia is by nutritional supplementation, including proteins, amino acids, Omega-3 fatty acids, and phytochemicals. Limited by the scope and focus, we are unable to cover this topic in the current review. Interested readers are referred to several excellent review articles below [41,42,43].

In summary, protein turnover during sarcopenia is characterized by higher proteolysis via calpains and autophagy. These increases are driven in part by higher levels of oxidative stress. In contrast to other degradation pathways, UPS activity is stagnant or even decreased during aging, whereas proteolysis by caspases is insignificant. Compared to younger individuals, elderly individuals cannot effectively synthesize protein in response to anabolic stimuli. This may be caused in part by elevated ROS that promotes excessive Akt/mTOR oxidation and inactivation.

## 3. Mitochondrial Homeostasis

### 3.1. Mitochondrial Oxidant Production

Mitochondria play an important role not only in cellular energy production but also in regulating ROS generation and antioxidant defense against oxidative stress, a major contributor to sarcopenia. Here, we focus on the importance of H_2_O_2_, a reactive species linked to both oxidative stress and adaptive redox signaling [5]. Both in vitro and in vivo studies indicate that mitochondria are the major source of basal H_2_O_2_ production [44]. Mitochondrial production of H_2_O_2_ increases with age in mice [45,46], rats [47,48], and humans [49]. Increases in H_2_O_2_ concentration induce atrophy in cell culture models [50], and time-course analysis shows that this increase takes place prior to skeletal muscle atrophy [51]. Exogenous H_2_O_2_ also decreases mitochondrial membrane potential that precedes mitochondrial fragmentation, despite no changes in transcripts related to mitochondrial fusion or fission protein expression [52]. Acute and chronic progression of H_2_O_2_ concentration similar to the chronic, low-grade inflammation seen in aging induces muscle atrophy in C2C12 myocytes, probably through increased protein degradation [26]. According to a variety of experimental models, increases in H_2_O_2_ precede and in some cases induce alterations in skeletal muscle volume with aging, suggesting that this plays a role in the development of sarcopenia.

A common compensatory response to increased oxidative stress is the upregulation of cellular antioxidant defenses [5]. Changes in muscle antioxidant enzyme activity with aging have been observed for decades [53], with the majority of research showing increases in activity of SOD1 [45], SOD2 [54], catalase [45,54,55], and GPx [55]. Moreover, mRNA levels of SOD1 and SOD2 have been shown to increase in 24-month-old mice with muscle atrophy, suggesting a transcriptional activation [45]. However, the same study showed decreases in SOD2 and GPx protein contents in aging muscles [45]. It seems possible that aging influences gene expression of antioxidant enzymes at both transcriptional and post-transcriptional levels [56], which warrants further investigation. 

Two questions may arise as to the role of mitochondrial source of ROS in developing sarcopenia: (1) Are increased mtROS the cause of muscle atrophy, and (2) if so, can increased antioxidant defense protect against muscle loss? A recent study by Eshima et al. [56] has provided some insight. Using a mitochondrial catalase-overexpression mouse model, the authors showed that eliminating mtROS did not prevent muscle atrophy induced by hindlimb unloading. Additionally, acute antioxidant treatments did not rescue the temporary loss of force following fatiguing contractions in young or aged skeletal muscle [57,58]. On the contrary, chronic manipulation of antioxidant enzyme content or activity has yielded mixed results. For example, Umanskaya et al. [59] showed that mitochondrial catalase overexpression reduced the age-dependent loss of muscle mass and function through decreased mitochondrial ROS and improved calcium handling. Because the substrate for catalase is H_2_O_2_ and aging can alter H_2_O_2_ concentration and distribution in different compartments of muscle cells [56]_,_ these results provide strong support for the notion that H_2_O_2_ might be a mechanistic driver of sarcopenia. Interestingly, in the senescence-accelerated mice (SAM), catalase, SOD, and GPx activities severely decreased by the age of 24 weeks, along with increases in oxidative stress, but a loss of grip strength did not occur until 32 weeks of age [12]. These studies reveal that mitochondrial production of ROS and the resultant redox status change are intimately involved in the development of sarcopenia.

### 3.2. Mitochondrial Biogenesis and PGC-1α Signaling

Mitochondrial homeostasis is controlled by mitochondrial biogenesis and degradation partially governed by autophagy/mitophagy and fusion/fission dynamics [60,61,62]. Due to the critical role it plays in mitochondrial biogenesis, PGC-1α has long been suspected as an important factor in the development of sarcopenia [63,64]. Mitochondrial biogenesis decreases in aging muscle cells, driven by an age-related decline in PGC-1α expression and possibly by decreased SIRT1 and SIRT3 levels as well, causing acetylation and inactivation of PGC-1α [65,66]. PGC-1α not only regulates mitochondrial biogenesis but also modulates the crosstalk of signaling pathways of mitochondrial quality control in old age [67]. For example, PGC-1α controls the expression of Mfn2, a main player in fusion dynamics and mitophagy, and aging decreases Mfn2 levels [68]. PGC-1α also promotes the expression of SIRT3, thus deacetylating the key mitochondrial metabolic and antioxidant enzymes [69,70].

Most studies demonstrate that aged muscle has significantly lower levels of PGC-1α content, signaling activity, and associated functions [64,71]. However, whether aging per se or other age-related deteriorative factors cause the downregulation of PGC-1α gene expression is still difficult to conclude. Recently, the SAM model has shed some light on muscular aging research including sarcopenia [12]. SAM mice demonstrated clear downregulation of genes involved in mitochondrial biogenesis including PGC-1α, nuclear respiratory factor-1 (NRF-1), Tfam, and cytochrome c oxidase, as well as mitochondrial fusion proteins Mfn2 and Opa1. This suggests that declined PGC-1α signaling capacity may be programmed in mammalian life. Supporting evidence also comes from PGC-1α knockout studies, wherein deletion of PGC-1α resulted in sarcopenia and shorter lifespan [72]. 

On the other hand, PGC-1α overexpression results in amelioration of not only muscle atrophy and deterioration caused by immobilization [72,73,74] but also attenuation of age-related decline in mitochondrial protein and Tfam expression, mitochondrial metabolic function, and muscle atrophy. Yet, whether PGC-1α overexpression can attenuate loss of muscle mass in aging is still controversial. Although previous research consistently shows positive effects of PGC-1α in boosting mitochondrial biogenesis and improving age-associated muscle function, some authors showed that PGC-1α overexpression via electroporation did not affect muscle fiber size or muscle/body weight ratio in old mice [75]. Additionally, PGC-1α had no effect on atrogen-1 or MuRF1 levels or protein ubiquitination, suggesting that there is no direct connection between these two signaling pathways.

Other than PGC-1α, nuclear factor erythroid 2-related factor 2 (Nrf2) deficiency has recently been implicated in frailty and sarcopenia by impairing mitochondrial biogenesis and dynamics in aged muscle [76,77]. In Nrf2 KO mice, muscle oxidative function and performance were reduced compared to those in the mid-age and old wild-type animals, along with decreased PGC-1α, Tfam, Mfn1/2, and Opa1 levels. However, whether Nrf2 prevented muscle fiber atrophy was not conclusive. The authors attributed the protective role of Nrf2 to ameliorating antioxidant gene expression and decreasing oxidative stress, thereby improving mitochondrial biogenesis and homeostasis at old age.

### 3.3. Autophagy and Mitophagy

Autophagy is one of four major proteolytic pathways that take place in skeletal muscle cells. There are three types of autophagy: macroautophagy, microautophagy, and chaperone-mediated autophagy [78]. Targeted degradation of mitochondrion by autophagy is termed mitophagy. Autophagy is a normal and necessary process to maintain healthy cells. With insufficient autophagy, the cell accumulates dysfunctional and malformed proteins, whereas excessive autophagy promotes muscle wasting [79]. Until the last few years, autophagy was generally thought to decrease with age [80]. This was supported by evidence that knockout of atrogenes and other autophagy proteins could induce early sarcopenia, while overexpressing atrogenes prevented losses in muscle function and extended lifespan [81]. However, more recent experiments have challenged this proposition by measuring autophagic flux in aged muscle [82,83,84]. Measuring only protein content depicts a static snapshot of autophagy, while flux measurements clarify discrepancies in results and interpretation. The findings of the above three studies revealed that basal autophagy and mitophagy flux both increase with age, but that aging blunted increases in mitophagy flux in response to exercise [84]. Other reviewers have noted that elevated basal mitophagic flux may be insufficient to maintain a pool of healthy mitochondria, given that dysfunctional mitochondria accumulate with age [85]. It was further noted that the inability to upregulate mitophagic response to exercise stimulus may exacerbate the accumulation of dysfunctional organelles [86].

Although autophagy is regulated by multiple pathways, several lines of evidence indicate that ROS stimulates autophagy in skeletal muscle [87]. For example, oxidative stress caused by SOD1 mutation [88] or by the addition of H_2_O_2_ to C2C12 cells induces autophagy [89,90,91]. Removing oxidative stress with antioxidant treatments also attenuates autophagy [89,92]. However, a study in mdx mice showed that excessive NOX-2-dependent ROS production suppressed autophagy [93].

The mechanism by which increases in autophagy following elevated ROS appears to involve mTORC1 and AMPK. An active mTORC1 generally suppresses autophagy by inhibiting ULK1 activity, whereas AMPK accelerates autophagy by phosphorylating ULK1 and by inhibiting mTORC1 [87]. It has been shown that oxidative stress induced by muscle immobilization activates proteolytic pathways and inhibits mTOR [94]. Studies in C2C12 cells also demonstrated ROS-mediated autophagy due to AMPK activation [89]. Furthermore, recent studies showed that Nrf2 deficiency, which increases oxidative stress, increased autophagic flux and AMPK activation in a manner similar to aging [82]. Detailed research has revealed that the concentration of ROS may dictate the outcome of autophagic response in muscles. For example, Meijles et al. [95] found that low levels of ROS promoted mTOR phosphorylation in cardiomyocytes, while high ROS levels could deplete ATP and activate AMPK. This study did not assess autophagy, but the concentration-dependent, divergent responses to ROS highlighted the complication of this topic. The characteristic increase in ROS at old age may in part explain the increased autophagic flux in aged muscle.

ROS also stimulates mitophagy by interacting with mitochondrial inner-membrane anion channels and mitochondrial permeability transition pores. An increase in ROS release by the mitochondria decreases mitochondrial inner membrane potential, thus labeling the dysfunctional mitochondria for degradation by mitophagy [96]. In support of this scenario, reducing oxidative stress from H_2_O_2_ with adiponectin treatment has been shown to reduce mitophagy in C2C12 cells [97]. A recent study demonstrated that suppressing oxidative stress with phytochemical antioxidant apigenin alleviated muscle atrophy due to inhibition of hyperactive mitophagy and apoptosis in aged mice [98]. Furthermore, PGC-1α overexpression by local transfection was shown to effectively suppress the age-related increase in mitophagy protein expression in mice, suggesting crosstalk between mitochondrial biogenic and mitophagic signaling pathways [75].

Molecular mechanisms explaining the redox control of autophagy/mitophagy are hot topics in muscle biology. One potential target of altered redox signaling in aging is PTEN, a phosphatase that normally dephosphorylates mTORC1 and suppresses Akt/PKB signaling pathway [99]. Posttranslational modifications of PTEN includes oxidation of its Cys 124 to form a disulfide bond, thus inhibiting its activity, leading to enhanced PI3K/Akt/mTOR signaling [99]. Moreover, Kim et al. reported that during muscle differentiation, mitochondrial ROS can oxidize PTEN, thereby promoting phosphorylation of Ser 317 on ULK1, a change that induces autophagy [100].

In sum, autophagy and mitophagy are redox-sensitive pathways whose basal levels in skeletal muscle rise with age due to increased mitochondrial oxidative damage. However, aging may impair the ability of skeletal muscle to activate mitophagy in response to elevated oxidative stress, such as during muscle contraction.

## 4. AMPK Signaling

AMPK is a pivotal enzyme in skeletal muscle that responds to cellular energy status and regulates a variety of cellular functions, and its activity is known to change with aging [101]. While AMPK itself does not control muscle mass, it coordinates downstream effectors, such as SIRTs and FOXO, which are key regulators of muscle protein turnover. AMPK is a heterotrimeric kinase with a catalytic α subunit, and β and γ regulatory subunits, which can combine to make up to 12 unique AMPK isoforms [102]. AMPK is a redox-sensitive kinase, though some controversy exists as to whether it can be directly activated by ROS or if the activation is a result of an indirect, ROS-induced decrease in cellular ATP levels. There is strong support for activation of AMPK independent of changes in ATP concentration across a variety of stimuli, such as addition of exogenous H_2_O_2_ [103], endogenous increases in H_2_O_2_ [104], and hypoxia [105]. Modification of cysteine residues on AMPKα appears to play an important part in these manipulations [104,106]. While exposure of AMPK to H_2_O_2_ resulted in increased kinase activity, mutation of Cys 299 to alanine diminished the ability of H_2_O_2_ to activate the enzyme, and mutation of Cys 304 to alanine totally abolished the stimulating effect of H_2_O_2_ [107]. On the other hand, oxidation of Cys 130 and Cys 174 on AMPKα inhibits its activity and interferes with its activation by AMPK kinases (AMPKK) [106]. Reduction of Cys 130/Cys 174 was shown to be essential for activation of AMPK during energy starvation [106]. Rabinovitch et al. [108] showed that mitochondrial ROS activated AMPK, which triggered a PGC-1α-dependent antioxidant response that decreased mitochondrial ROS. Cells lacking AMPK activity displayed higher mitochondrial ROS and premature senescence. These findings clearly demonstrate that in addition to being sensitive to energy status, AMPK is redox-sensitive and mitochondrial ROS regulates AMPK activity via its α-subunit cysteine redox regulation.

Based on the fact that AMPK is redox-sensitive, one would expect AMPK activity to increase in the presence of increased ROS with aging. This would be important for further research on sarcopenia, as increased basal activation of AMPK has been shown to diminish muscle overload-induced hypertrophy in both young and old muscle fibers [109]. However, AMPK activity has been shown to decrease with age in rat and human skeletal muscle, despite maintenance of its protein content [110,111]. Interestingly enough, combined treadmill running and resistance training or aerobic training alone in rodents were able to attenuate sarcopenia in part by upregulating AMPK and PGC-1α [112,113]. 

Understanding the mechanisms of the interaction between AMPK and aging is crucial for elucidating the prevention and treatment of sarcopenia. Several potential pathways have recently been explored. During the past two decades, the double-negative feedback loop between mTORC1 and AMPK was postulated to explain the enhanced autophagy under oxidative stress [114,115]. This theory claims that under physiological conditions, mTORC1 inhibits AMPK and thus keeps ULK1, the initiator of autophagy, in check. An increase in AMPK activity under oxidative stress and nutritional deficiency not only downregulates mTORC1, the master regulator of protein synthesis, but also removes the negative feedback inhibition on AMPK. Hyperactivation of AMPK can phosphorylate ULK1, leading to increased autophagic flux. In our analysis, this double-negative theory may account in part for the age-associated decline in protein synthesis as well as increased proteolysis via autophagy. Another potential explanation is attributed to redox resistance within AMPK, supported by a recent study by Caldeira et al. [116]. In this study, AMPK from peripheral blood mononuclear cells of middle-aged individuals was able to be activated in response to an oxidative challenge and reduce oxidative stress. In contrast, the same stimulus did not induce an antioxidant response in cells from elderly individuals. Recently, Nrf2-mediated downregulation of AMPK was studied in an attempt to elucidate the crosstalk between these two signaling pathways in autophagy under oxidative stress [117]. Nrf2 has a fundamental role in cell homeostasis by regulating antioxidant and electrophilic stress responses. While prolonged oxidative stress, as is seen in sarcopenia, increases AMPK activity and content, it also activates Nrf2 and subsequently modulates AMPK activity and autophagy. It was suspected that this reduction in AMPK prevents hyper-autophagic responses. While this work was performed in HEK293T cells and *C. elegans*, it may shed some light on the role of AMPK and autophagy in aging and sarcopenia. 

The decrease in AMPK detailed above is distinct from the increase in AMPK activity seen in other muscle-wasting conditions such as immobilization. In those conditions, myocytes enter a low energy state, upregulating AMPK activity [62]. This inhibits the PI3K/Akt/mTOR pathway and upregulates FOXO-mediated transcription. The net result is accelerated autophagy and increased degradation, primarily by MuRF1, atrogin-1, and several atrogenes [62]. However, as stated above, AMPK and FOXO activities do not increase with age. Instead, sarcopenia is attenuated through increases in AMPK activity. As stated in the Mitochondrial Biogenesis section, this is likely mediated by AMPK phosphorylation of PGC-1α. We and others have demonstrated exercise to be one of the most successful means to increase AMPK activity and mitigate sarcopenia [65,112,118]. The divergent responses of AMPK to these conditions highlight the multi-functional nature of this kinase. Excessive AMPK activity may be relevant to sarcopenic individuals requiring immobilization or bedrest, and how these stressors together affect AMPK, FOXO, and atrophy requires further research.

## 5. Apoptosis and Regeneration

One of the molecular mechanisms of sarcopenia is postulated to be an imbalance when apoptosis outpaces regeneration [16]. Indeed, the imbalance comes from both directions: myonuclear apoptosis increases [119] while satellite cell function decreases [29]. There are two main apoptotic pathways. The first is the intrinsic, mitochondria-dependent pathway in which mitochondria release cytochrome *c* and promote cell death [120]. This pathway is accelerated by elevated concentrations of intracellular calcium ions [121] and ROS [122]. The extrinsic pathway relies on death receptor binding, such as TNFα, and directly stimulates apoptosis or induces the intrinsic pathway. TNFα binds to membrane-bound receptors such as tumor necrosis factor receptor 1 (TNFR1), which recruits adapter proteins such as tumor necrosis factor receptor type 1-associated DEATH domain (TRADD) that eventually activate caspase-3. For details on the role of TNFα and apoptosis, readers are referred to other reviews [119,123,124]. Currently, it is unclear whether the intrinsic or the extrinsic pathway is the major player in sarcopenic apoptosis [29].

### 5.1. Apoptosis

In the intrinsic pathway of apoptosis, elevated ROS leads to higher intracellular Ca^2+^ concentrations mediated in part by redox modifications to the ryanodine receptor that promote Ca^2+^ influx into the cell [17]. This phenomenon is also observed in hypobaric hypoxia, another condition that elevates ROS and promotes muscle atrophy. Briefly, oxidative stress hypernitrosylates the ryanodine receptor and impairs binding with FKBP12/calstabin-1 and other channel complexes [17]. Furthermore, elevated ROS due to SOD2 knockout [19], chronic stimulation [20], and aging [18] can inhibit the activity of SERCA pumps and lead to higher intracellular Ca^2+^ levels, which increases the permeability of mitochondria, release of cytochrome *c*, and activation of caspase-3 [119,121]. Furthermore, calpain activity stimulated by higher intracellular Ca^2+^ induces apoptosis by activating caspase-12 and prompting pro-apoptotic factor endonuclease G (EndoG) release from the mitochondria [13,125]. 

As mentioned above, TNFα is a prominent pro-apoptotic ligand that can initiate the extrinsic apoptotic pathway by binding with its receptors. Moreover, TNFα binding stimulates ROS production, inducing mitochondria-dependent apoptosis [126]. Importantly, TNFα levels increase with age in both rats and humans, and TNFα levels are inversely proportional to muscle force production [127]. These findings strongly suggest that TNFα is one of the mechanistic drivers of sarcopenia, and it has been postulated as a key player in the inflammatory theory of aging [1,38].

### 5.2. Satellite Cells and Regeneration

The ability of satellite cells (SC) to participate in muscle regeneration declines with age, and this process is partially mediated by elevated oxidative stress. Specifically, SC differentiation and proliferation decrease with muscle aging [29], even in response to anabolic stimuli [128]. Additionally, in aged muscles, more SCs enter the permanent senescence stage from reversible quiescence [129]. Furthermore, antioxidant capacity in SC diminishes with age [130], whereas ROS production increases with age [131]. In light of the role of redox balance within the SC, antioxidant supplementation was found to renew the regeneration capability and improve antioxidant defense in human senescent SCs [132]. In addition to myocytes, there are ROS-dependent alterations in autophagy in SCs [132]. Basal autophagy flux within SCs declines with age, and this contributes to their senescence and dysfunction. While geriatric SCs produced more ROS, inhibiting ROS restores the proliferation and regeneration capacity of SCs in geriatric cells [132]. Finally, a reduced as opposed to a more oxidized cellular environment promotes myoblast differentiation in C2C12 cells [133]. These findings demonstrate the importance of redox environment within SCs to preserve regeneration and delay senescence.

There is clear evidence that elevated ROS from aging negatively affects the regeneration ability of SC. However, precise mechanisms underlying the observations are still unknown. Changes in several redox-sensitive signaling pathways in myogenesis such as Notch, Wnt, p38/MAPK, and JAK/STAT3 have been observed in satellite cells [134]. It remains unclear why the responses of these redox-sensitive pathways change with age and why they are no longer adaptive as the organism ages.

## 6. Inflammation

The impact of inflammation on almost all aging-related degenerative diseases including sarcopenia has been increasingly recognized by the scientific community in the last decade, and this association clearly hinges on cell signaling [135]. Several cytokines play an important role in the pathogenesis of muscle wasting, most notably TNF-α, but also IL-1β, IL-6, interferon (IFN)-γ, and transforming growth factor (TFG)-β [136]. Inflammation represents a pathophysiological state that is intimately related to cellular redox stress and in turn substantially alters cellular redox homeostasis. Inflammation has long been considered a major threat during muscle disuse for various reasons [137,138]. The main trigger of inflammation in immobilized muscle is activation of the NF-κB pathway. Transgenic mice overexpressing IKKβ, the primary activator of NF-κB, suffer from muscle wasting via the upregulation of MuRF1 [136]. In contrast, deletion of p105/p50 subunits of NF-κB was shown to be resistant to IM-induced muscle atrophy [139]. NF-κB activation is mostly associated with an overproduction of pro-inflammatory cytokines, as mentioned above. TNFα, the most potent activator of NF-κB pathway, promotes a positive feedback loop by activating NF-κB, which in turn induces TNFα and drives NF-κB mediated muscle wasting. TNFα, IL-1, and IL-6 directly inhibit PI3K/Akt activity and thus release their inhibition on FOXO [140]. NF-κB also promotes the expression of MuRF1, which degrades muscle contractile proteins [136]. Furthermore, inflammation is associated with an overexpression of myostatin due to FOXO activation and can induce muscle atrophy independently of NF-κB [141]. Thus, muscle inflammation can be a sustained disturbance to muscle redox homeostasis due to the crosstalk of multiple redox signaling pathways that control protein turnover.

Recent research indicates that PGC-1α has an anti-inflammatory effect in skeletal muscle atrophy. PGC-1α modulates local or systemic inflammation by suppressing the expression of pro-inflammatory cytokines and myokines such as TNF-α and IL-6 [142]. The inhibitory effects of PGC-1α on inflammation may result from several cellular mechanisms. First, intact PGC-1α signaling is mandatory for antioxidant enzyme gene expression [143]. Second, PGC-1α inhibits NF-κB by interaction with the p65 subunit in the nucleus, thus attenuating its binding to DNA [144]. Third, PGC-1α suppresses nuclear retention of FOXO3 caused by AMPK activation [145]. Finally, SIRT expression is at least partially controlled by PGC-1α, and SIRT1 negatively regulates FOXO and p65 via their deacetylation [10]. However, although overexpression of PGC-1α via in vivo transfection was found to ameliorate mitochondrial function and reduce upregulated mitophagy and ubiquitin proteolysis in aged muscle, it failed to prevent age-associated muscle atrophy [75]. 

Convincing evidence exists linking inflammation with muscle metabolic dysfunctions marked by disturbed mitochondrial homeostasis, insulin resistance, and protein imbalance. Especially, mitochondrial dysfunction is recognized as a major factor of age-related muscle degeneration [146]. However, the molecular mechanism connecting sarcopenia to inflammation is poorly understood. Studies based on large databases have generated a more complicated picture and controversial conclusions. For example, one recent study using meta-analysis reported that, independently of disease state, higher levels of C-reactive protein (CRP), IL-6, and TNFα were associated with lower arm and leg strength and muscle mass [147]. Yet another meta-analysis study, while finding that sarcopenic subjects had significantly higher levels of CRP than control subjects, showed that serum IL-6 and TNFα levels were not significantly different when people with sarcopenia were compared with controls [148].

However, NF-κB alone does not appear to be a driver of sarcopenia. p65 protein expression and NF-κB binding did not increase in aged rats despite elevations in TNFα [149], suggesting a disconnect between NF-κB and its typical stimuli with age. Rather, the negative impact of NF-κB on sarcopenia is due to impaired adaptation to anabolic stimuli. Specifically, elderly individuals showed increased nuclear Ser468 phosphorylation of p65, a DNA binding region of NF-κB, in response to a bout of resistance exercise. In comparison, young participants saw no change in p65 phosphorylation, suggesting an impaired NF-κB response to exercise with age [150]. This study also demonstrated reduced phosphorylation of S6K1 and FOXO with exercise, again highlighting the role of anabolic resistance in the elderly. In aged rats, impairing inflammatory cytokines with ibuprofen improves acute Akt phosphorylation of FOXO3 after feeding [151]. Although NF-κB proteins were not measured, minimizing low-grade inflammation may mitigate anabolic resistance and therefore improve outcomes in sarcopenia. Despite theoretical similarities to immobilization, the role of inflammation in sarcopenia may be related more to anabolic resistance than to direct protein degradation. 

## 7. Neuromuscular Junction

Degradation of the nervous system with age is well established [1]. Defects of neuromuscular function not only have negative impacts on muscle contractile function but also result in oxidative stress, as denervation increases mitochondrial peroxide production in neighboring, innervated muscle fibers [152]. Much of the mechanistic understanding of the changes in neuromuscular function with age centers on the use of SOD1 KO mice. Complete SOD1 KO induces a sarcopenic phenotype, characterized by a loss of strength, muscle mass, and reduced physical activity [153]. Importantly, muscle-specific SOD1 KO can lead to clear reductions in contractile force but little to no loss in muscle mass [154,155]. Normal expression of SOD1 in nervous tissue is critical, as neuron-specific SOD1 KO resulted in contractile force loss in muscle [156], while restoration of strength and mass was observed when this line of SOD1 was restored [157]. The disconnect between muscle mass and function is due to the decrease in excitation–contraction coupling, specifically mediated by impaired SERCA pump function and damage to the ryanodine receptor [158]. Recently, a SERCA activator was shown to reverse the alterations in muscle mass, function, oxidative damage, and autophagy markers back to wild-type levels in SOD1 KO mice [159]. 

The above-mentioned tissue-specific SOD1 KO model reveals that the signaling mechanism of the neuromuscular junction (NMJ) is probably distinct from the skeletal muscle fiber atrophy. Neuron-specific SOD1 KO induced progressive NMJ degeneration and gradual loss of strength and muscle mass, which was most pronounced at 24 months of age. This occurred despite no alterations in mitochondrial oxidative damage or function [160]. However, muscle-specific SOD1 KO does not lead to degradation at the NMJ or alterations in redox homeostasis, which are both found in a whole-body SOD1 KO model [154]. Thus, the results of SOD1 KO mice can be synthesized into a two-hit model, where both loss of redox homeostasis in motor neurons and increased skeletal muscle ROS are necessary to induce the sarcopenic phenotype seen in whole-body SOD1 KO [161]. These findings indicate that dysregulation in redox homeostasis within motor neurons is a major contributor to sarcopenia [154,155,156,157].

Recent evidence suggests that signaling can also occur in a retrograde fashion, i.e., from the muscle cells back to the innervating nerves. This scenario is best supported by a series of SOD1 mutant studies, initially designed for the study of amyotrophic lateral sclerosis (ALS) [162]. Mice with a muscle-specific SOD1 mutation (SOD1^G93A^) develop fiber atrophy, loss of strength, and mitochondrial dysfunction [88], making it a possible model for sarcopenia. Expression of SOD1^G93A^ specifically in skeletal muscle lead to disruptions in NMJ structure, improper mitochondrial localization, and impaired transmembrane potential in pre- and post-synaptic mitochondria [163]. It has been shown that transgenic expression of PGC-1α caused morphological and functional remodeling, including improved NMJ, providing further evidence for retrograde signaling from skeletal muscle back to the NMJ [164]. 

Several studies have attempted to further understand the mechanisms behind muscle-driven or nerve-driven disorders of signaling in sarcopenia. Comparison of full SOD1 KO vs. muscle-specific SOD1 KO showed divergent oxidation statuses of the catalytic site of Prx5 [165]. Despite these changes in Prx, the peripheral nerves of the whole-body SOD1 KO mice did not show any evidence of oxidative damage. This is consistent with previous research showing that the catalytic cysteine residue of Prx6 in peripheral nerves was oxidized in sarcopenic mice, alongside increases in superoxide and peroxynitrite production [165]. Oxidation of Prx leads to their inactivation [166], impairing their ability to form so-called redox relays with downstream effectors [167]. This implies that Prxs of the peripheral nerves in sarcopenic mice are unable to effectively transmit the increase in H_2_O_2_ into adaptive signaling, leading to increased oxidative stress and cellular dysfunction. Overall, these data suggest that manipulation of signaling due to changes in the redox status of cysteine residues, particularly those on enzymes responsible for redox signal transduction, instead of the oxidative damage itself, could be a potential mechanistic driver of sarcopenia.

Nevertheless, oxidative damage should not be totally discounted in the development of sarcopenia, particularly within the mitochondria. SOD1 mutation in mice was shown to induce loss of mitochondrial inner membrane (MIM) potential near the NMJ, which interfered with propagating Ca^2+^ waves and signaling in skeletal muscle [168]. This defect of neuromuscular coupling was proposed to underlie the pathogenesis of ALS-induced muscle atrophy. On the other hand, transgenic expression of SOD1 in the mitochondrial intermembrane space was able to reduce mitochondrial oxidative damage, allow motor neurons to propagate, prevent NMJ damage, and eliminate the loss of muscle mass and function in SOD1 KO mice [169]. Furthermore, in SOD1 mutant mice developing sarcopenia, treatment with Trolox was able to rescue mitochondrial function and subcellular organization, restore NMJ morphology, and return acetylcholine receptor turnover to age-matched WT levels [163]. The same study also examined the role of PKC_ϴ_, which is a redox-sensitive kinase expressed in the post-synaptic region of the NMJ. PKC_ϴ_ serves to eliminate acetylcholine receptors in skeletal muscle and mediate nerve-muscle interactions [170,171]. Treatment of SOD1 mutant mice with a PKC_ϴ_ inhibitor rescued loss of mass and force [163]. This implies that PKC_ϴ_-mediated oxidative stress may activate downstream pathways related to NMJ degradation. Because treatment with Trolox produced similar results to the PKC_ϴ_ inhibition, available evidence suggests a role for redox signaling in the degradation of the NMJ in sarcopenia. Taken together, research up to this date highlights the importance of maintaining redox signaling in muscle neurons and the NMJ in the prevention of sarcopenia. 

## 8. NAD^+^ Homeostasis

Increased protein acetylation plays an important role in the mitochondrial and functional decline in aged muscle and can be a potential mechanism for sarcopenia [172]. Over 100 lysine sites in mitochondrial proteins may be acetylated, which may alter mitochondrial function in an adverse manner. This can include the inactivation of PGC-1α and key enzymes in the TCA cycle and ETC, inactivation of SOD2 and other antioxidant enzymes, and activation of NF-κB and inflammatory pathways [173,174,175]. Thus, maintenance of cellular protein acetylation/deacetylation balance is critical for maintaining functional integrity in aging muscle.

Protein deacetylation is carried out mainly by the NAD^+^-dependent deacetylase SIRTs [176]. Previous studies have shown that aging downregulates SIRT1 activity, the multi-functional deacetylase in the cell; however, recent research indicates that SIRT1 protein levels are not altered with age and some studies even reported positive correlations between SIRT content and age in muscle despite elevated acetylation of protein content [177]. Thus, cellular SIRT levels per se do not appear to determine protein acetylation/deacetylation status in skeletal muscle. Instead, it is suggested that the cellular content of NAD^+^, the substrate and mandatory acceptor of lysine residues from acetylated protein in SIRT-catalyzed reactions, is significantly downregulated during aging in general [172,176,177] and in skeletal muscle in particular [178]. Thus, the insufficient supply of NAD^+^ may be a limiting factor in intracellular deacetylation capacity in aged muscle, because deteriorated mitochondrial function and inflammation are both important etiological mechanisms for sarcopenia [179].

Cellular mechanisms by which aging diminishes the cellular NAD^+^ pool are complicated and under intense research. While the metabolic pathways that use NAD^+^ and NADH as coenzymes to transfer electrons do not result in net NAD^+^ decline in the cell, several signaling pathways are known to consume NAD^+^, such as SIRTs, poly(ADP-ribose) polymerase (PARP)-1, and a cluster of differentiation (CD) 38 [176,177]. During aging, each of these pathways is activated and competes for NAD^+^. For example, aging induces mitochondrial protein hyperacetylation in muscle, requiring increasing amounts of NAD^+^ in the TCA cycle, ETC, and SOD2 [172]. Activation of inflammatory pathways during aging demands NAD^+^ for deacetylating NF-κB-p65 and FOXO in the nucleus [176]. Furthermore, aging is known to increase DNA oxidative damage [180]. Repairing oxidized DNA base pairs uses NAD^+^ as a substrate to transfer poly ADP-ribose backbone, catalyzed by PARP-1. Research shows that up to 80% of NAD^+^ decline during aging may be explained by increased DNA damage [176]. Indeed, PARP-1 was upregulated in aged mouse muscle [178]. Interestingly, PARP-1 inhibition was shown to improve mitochondrial function due to SIRT1 activation, presumably due to increased NAD^+^ availability [181]. Finally, aging increases protein expression of CD38, also known as cyclic ADP ribose hydrolase, a multi-functional enzyme that produces ADP ribose and cyclic ADP-ribose (minor function) from NAD^+^ [182]. Inhibition of CD38 or CD38 gene knockout has been found to preserve intracellular NAD^+^ levels and SIRTs activity [183,184]. CD38 protein content was reported to increase dramatically in aged mouse skeletal muscle, along with decreased mitochondrial oxidative enzyme expression, increased protein and transcription factor acetylation, and greater oxidative damage [178]. 

Under physiological conditions, decreased cellular NAD^+^ concentration may be replenished by elevated NAD^+^ synthesis by either the Preiss-Handler pathway that converts dietary nicotinic acid (NA) to NAD^+^ or the de novo NAD^+^ biosynthesis from dietary tryptophan [176]. Whether the enzyme activities in the Preiss–Handler pathway and de novo biosynthesis pathway are downregulated or upregulated in aging muscle is unclear. More importantly, NAD^+^ levels may be restored by the so-called salvage pathway controlled by nicotinamide phosphoribosyltransferase (NAMPT), the rate-limiting enzyme-converting nicotinamide (NAM) generated by SIRTs, CD38, and PARP-1 to nicotinamide mononucleotide (NMN) and eventually NAD^+^. This pathway recycles NAM to maintain intracellular NAD^+^ levels and relieves NAM inhibition on SIRTs [176]. Aging may decrease NAMPT activity, thus reducing the conversion of NAM to NAD^+^, resulting in muscle degeneration [185,186]. However, this area is still controversial as Yeo et al. [178] showed an upregulation of NAMPT in quadriceps and gastrocnemius muscles in aged mice, yet with decreased NAD^+^ levels. Dietary intake of NAD^+^ precursors could have a strong influence on NAD^+^ homeostasis in aged animals, as some recent research indicates that supplementation of NAM, or nicotinamide ribose (NR) can boost intracellular and mitochondrial NAD^+^ levels, improve metabolic function, and even increase longevity in aged mice [177].

In summary, aged muscles suffer from NAD^+^ deficit due to increased competition from SIRTs, CD38, and PARP-1 pathways, whereas neither de novo synthesis nor NAD^+^ salvage is sufficient to replenish the intracellular NAD^+^ pool. The shortage of NAD^+^ may retard the ability of SIRT to deacetylate key enzymes and transcription factors. Whether or not NAD^+^ deficiency constitutes a major mechanism for sarcopenia requires more investigation.

## 9. Peroxiredoxins and Redox Signaling 

Since the free radical theory of aging was postulated more than half a century ago, research on the mechanisms of aging has been one of the most vibrant branches of modern biology. Sarcopenia is proven to underlie numerous health hazards beyond muscle itself. The connection of increased ROS with age and sarcopenia does not seem to be coincidental. Most age-related alterations reviewed in this article are linked with an imbalance between ROS production and antioxidant defense, favoring an oxidized cellular milieu. While some of the enzymes and transcription factors are directly modified by H_2_O_2_ and other reactive species_,_ others show sensitivity to intracellular redox status that either change their gene expressions or undergo posttranslational modifications. Aged muscle appears to lose some adaptive response to redox signaling normally seen in younger muscles. Thus, there has been an intense search for the key factors causing this age-related decline in adaptability. 

Prx has recently emerged as one of the potential and promising explanations for dysregulated redox signaling in aging skeletal muscle [187]. Redox modulation of cysteine residues is a common mechanism in redox signaling, and Prxs are enzymes readily subject to cysteine oxidation by H_2_O_2_ [188]. Prxs can “exchange” these oxidized thiol groups with reduced thiols on redox-sensitive proteins [189]. This oxidation is usually reversible, but Prxs can become “stuck” in an irreversible oxidized state that can no longer transfer their disulfide bonds to other redox-sensitive proteins [3]. This is one hypothesis that may explain the diminishing adaptive response of skeletal muscle to conditions that promote oxidative stress, such as during exercise [187].

Some members of the Prx family (Prx1-5) act as downstream signaling mediators for several key signaling pathways controlled by MAP3Ks, such as apoptosis signal-regulating kinase 1 [190], MEKK4 [191], and PTEN [192]. Prxs are unique compared to other antioxidant enzymes due to their high reactivity with H_2_O_2_, which itself does not always oxidize downstream targets, because of its relatively low physiological concentrations (< 100 nM) in muscle cells, even after an exercise stimulus [189]. Prxs have been shown to mediate the oxidative signal from H_2_O_2_ to target transcription factors such as PPARγ and NF-κB [187]. Overexpression of Prx3 was shown to abolish myocyte atrophy, loss of contractile force, and ROS production in a SOD-1 KO mouse model [193], whereas Prx6 KO has been shown to induce muscle atrophy via an increase in MuRF1, a primary mediator of ubiquitin proteolysis [194]. Furthermore, Prx3 is heavily involved in ameliorating mitochondrial homeostasis and muscle contractile function [195]. 

There is considerable evidence that Prxs may be the key mediator to convey redox signals to contracting muscle cells during exercise [187,189]. Prx content remains stable from adulthood to old age in mice [189]. Although training was found to increase basal Prx5 content in elderly men above that in young men, Prx5 content decreased after acute exercise in the elderly, whereas exercise did not elicit such a reduction in young subjects [196]. Moreover, Prx2 oxidation in muscle fibers in old mice was diminished following contractile activity compared to young mice [189]. When Prx3 was overexpressed in a sarcopenic model induced by SOD1 KO, it rescued the adverse effects caused by a lack of antioxidant protection [193]. These studies indicate that the response of Prx to exercise may decline with old age and the activation of Prx by H_2_O_2_ diminishes in aged muscle. The role of Prx dynamics during aging and exercise is an interesting and promising area for future research on sarcopenia.

## 10. Conclusions

Despite decades of research that have identified multiple pathophysiological factors contributing to age-related muscle deterioration, the primary mechanism that could pinpoint and unlock this health-threatening problem in the elderly is still elusive. However, recent research suggests that age-associated dysregulation of redox signaling may underlie many deteriorations seen at physiological, subcellular, and molecular levels. A discord of mitochondrial homeostasis with aging affects most observed problems and requires attention. Furthermore, age-related alteration of Prxs appears to provide a key link between declined redox signaling and the functional deficit in aging. The search for the primary suspect of the fundamental mechanisms for sarcopenia will likely continue. 

## Figures and Tables

**Figure 1 ijms-22-09045-f001:**
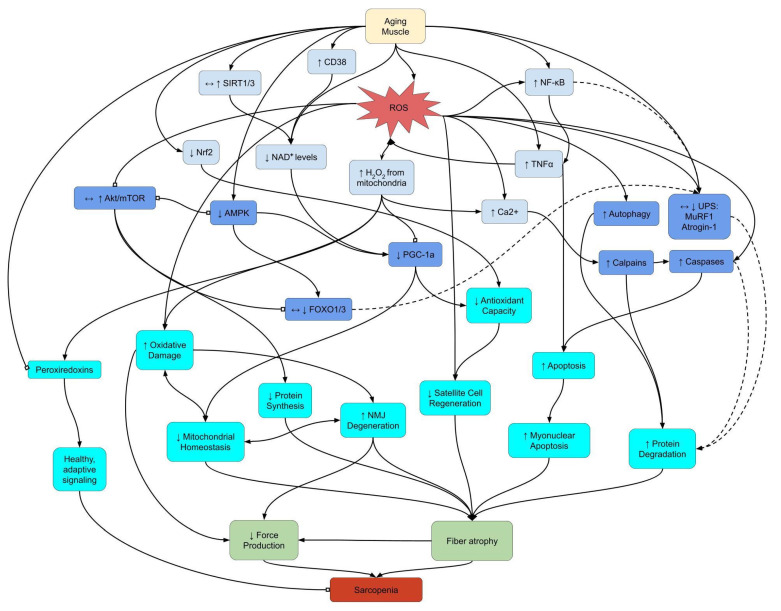
Select redox signaling pathways underlying sarcopenia. Lines between boxes indicate signaling pathway connections. Arrowheads represent an increase or promotion, while block heads represent inhibition. Dashed lines indicate pathways with a minor or negligible role in sarcopenia. Arrows within boxes show changes in sarcopenia based on upstream signals. Abbreviations: AMPK: 5′ adenosine monophosphate-activated protein kinase; Ca^2+^: calcium ions; CD38: cluster of differentiation 38; H_2_O_2_: hydrogen peroxide; mTOR: mechanistic target of rapamyacin; NAD^+^: nicotinamide adenine dinucleotide; NF-κB: nuclear factor kappa-light-chain-enhancer of activated B cells; NMJ: neuromuscular junction; Nrf2: nuclear factor erythroid 2-related factor 2; PGC-1α: peroxisome proliferator-activated receptor gamma coactivator 1-alpha; ROS: reactive oxygen species; SIRT: sirtuin; TNFα: tumor necrosis factor alpha; UPS: ubiquitin-proteosome system.

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
