# Peer review of "Redox Signaling and Sarcopenia: Searching for the Primary Suspect"

_ijms, 2021, doi:10.3390/ijms22169045_

Round 1

Reviewer 1 Report

This review article reported that age-related alteration of Prxs appears to provide a key link between declined redox signaling and the functional deficit in aging. This review integrates knowledge of oxidative stress, mitochondria function, and sarcopenia. A revision is suggested.

  1. Fig1, Sirtuins are a family of signaling proteins involved in metabolic regulation. Please clarify which one that authors want to reveal? Also, please discuss the function between SIRT and mitochondria function and muscle wasting.
  2. Please address how aging regulate pro-inflammatory responses, thereby mediating sarcopenia.
  3. Please discuss mitochondria biogenesis in this model.
  4. Please discuss the prevention of aging-mediated sarcopenia. Any strategy or intervention?
  5. “8. Peroxiredoxins and Redox Signaling “ lacks of linkage with above content.
  6. I suggest authors should focus on discussing AMPK/oxidative stress/SIRT/ mitochondria function.

Reviewer 2 Report

Redox Signaling and Sarcopenia: Searching for the Primary Suspect

Foreman, Hesse, and Ji – IJMS

The authors present a thorough review of the literature exploring the molecular pathways involved in redox signaling in skeletal muscle. The authors use this review to inform on the current state of the field regarding the contribution of these signaling pathways to the development of sarcopenia.  This is a well written and organized review which has no major flaws that would prevent its publication after minor grammatical and technical modifications to the text.  Several specific suggestions are listed below:

Line 27 – Are there references which indicate what the estimated overall health care cost is?  Reporting and citing this information would be helpful.

Line 47 – A slight expansion on Peroxiredoxin would be helpful here, even if it is just mentioned that it is a peroxide scavenger.

Line 76 – C2C12 is not commonly expressed with the numbers as subscripts.  This makes it look like a chemical and is confusing.  Please consider revising throughout.

Line 102 – It is contradictory to suggest that the mTOR pathway has been discussed in many reviews and yet to only cite one, which is more focused on sarcopenia than on mTOR/AKT.

Line 149 – Calpains, Caspases, and Autophagy were discussed in lines 73-98. The parenthetical suggestion that it will be discussed later is misleading here and can be removed.

Line 178 – The citation for #39 here is in a different format. Please change.

Line 283 – Should this read “…by interacting with the mitochondrial inner-membrane…”?

Line 293 – Should this read “pathways”?

Line 294 – Should the read “molecular mechanisms”?

Line 348 – This statement is a well formed supposition based upon solid facts and observations from previous studies, making it more of a “speculation” rather than an “opinion”.

Line 422 – Should this read “negative impacts”?

Line 460 – Should this read “to further understand”?

Line 591 – Please change the “y” to a “γ” in Ppary.

Reviewer 3 Report

The Review “Redox Signaling and Sarcopenia: Searching for the Primary 

Suspect” by Foreman and coworkers, refers to the association between redox signaling and sarcopenia. The Review has been nicely performed, but minor revision is required in particular to remove a few typos mistakes (I.e line 24, 83, 160, 164, 178, 181, 198, 264, 266, 29, 368,392, 522, 525, 532, 540, etc; on line 170 add some commas, from line 564 to 575 add some references). Concerning the figure, I suggest to put close both title and figure description. In addition, I suggest  to firstly introduce the chemical formula with the name of the  compound. Moreover, I wondered if a paragraph on the implication of amino acids and amino acid sensing in aging and diseases could fit in this review. To this aim I suggest you the paper by Dato et al (Biogerontology (2019) 20:17–31 https://doi.org/10.1007/s10522-018-9770-8(0123456789().,-volV()0123456789().,-volV) ). In addition, I guess that in the proximity of line 102 some references must be added. After this minor revision, I consider the paper ready for publication.

Round 2

Reviewer 1 Report

My questions had been addressed, accept in present form is suggested